# Applicability of Diagnostic Criteria and High Prevalence of Familial Dysbetalipoproteinemia in Russia: A Pilot Study

**DOI:** 10.3390/ijms241713159

**Published:** 2023-08-24

**Authors:** Anastasia V. Blokhina, Alexandra I. Ershova, Anna V. Kiseleva, Evgeniia A. Sotnikova, Anastasia A. Zharikova, Marija Zaicenoka, Yuri V. Vyatkin, Vasily E. Ramensky, Vladimir A. Kutsenko, Svetlana A. Shalnova, Alexey N. Meshkov, Oxana M. Drapkina

**Affiliations:** 1National Medical Research Center for Therapy and Preventive Medicine, Ministry of Healthcare of the Russian Federation, Petroverigsky per. 10, Bld. 3, 101000 Moscow, Russia; alersh@mail.ru (A.I.E.); sanyutabe@gmail.com (A.V.K.); sotnikova.evgeniya@gmail.com (E.A.S.); azharikova89@gmail.com (A.A.Z.); vyatkin@gmail.com (Y.V.V.); ramensky@gmail.com (V.E.R.); vlakutsenko@yandex.ru (V.A.K.); sshalnova@gnicpm.ru (S.A.S.); meshkov@lipidclinic.ru (A.N.M.); drapkina@bk.ru (O.M.D.); 2Faculty of Bioengineering and Bioinformatics, Lomonosov Moscow State University, 1-73, Leninskie Gory, 119991 Moscow, Russia; 3Phystech School of Biological and Medical Physics, Moscow Institute of Physics and Technology, Institutskiy per. 9, 141701 Dolgoprudny, Russia; marija.zaicenoka@gmail.com; 4Department of Natural Sciences, Novosibirsk State University, 1, Pirogova Str., 630090 Novosibirsk, Russia; 5Faculty of Mechanics and Mathematics, Lomonosov Moscow State University, 1-73, Leninskie Gory, 119991 Moscow, Russia; 6National Medical Research Center for Cardiology, 3–ya Cherepkovskaya Street, 15A, 121552 Moscow, Russia; 7Research Centre for Medical Genetics, 1 Moskvorechye St, 115522 Moscow, Russia; 8Department of General and Medical Genetics, Pirogov Russian National Research Medical University, 1 Ostrovityanova st., 117997 Moscow, Russia

**Keywords:** familial dysbetalipoproteinemia, hyperlipoproteinemia type III, *APOE*, apolipoprotein E, apolipoprotein B, autosomal dominant, remnant lipoproteins, hyperlipidemia, population, genetic

## Abstract

Familial dysbetalipoproteinemia (FD) is a highly atherogenic genetically based lipid disorder with an underestimated actual prevalence. In recent years, several biochemical algorithms have been developed to diagnose FD using available laboratory tests. The practical applicability of FD diagnostic criteria and the prevalence of FD in Russia have not been previously assessed. We demonstrated that the diagnostic algorithms of FD, including the diagnostic apoB levels, require correction, taking into account the distribution of apoB levels in the population. At the same time, a triglycerides cutoff ≥ 1.5 mmol/L may be a useful tool in identifying subjects with FD. In this study, a high prevalence of FD was detected: 0.67% (one in 150) based on the ε2ε2 haplotype and triglycerides levels ≥ 1.5 mmol/L. We also analyzed the presence and pathogenicity of *APOE* variants associated with autosomal dominant FD in a large research sample.

## 1. Introduction

Familial dysbetalipoproteinemia (FD), also known as hyperlipoproteinemia type III (OMIM #617347), is a genetically based lipid disorder caused by variants in the *APOE* gene. FD is associated with elevated levels of lipoprotein remnants and an increased risk of premature atherosclerosis [1,2].

*APOE* has three main alleles that encode apolipoprotein E isoforms: ε2, ε3, and ε4. As a result, there are three homozygous (ε2ε2, ε3ε3, and ε4ε4) and three heterozygous (ε2ε3, ε3ε4, and ε2ε4) haplotypes. The haplotype composition is determined by the combination of two variants: rs7412 (NP_000032.1:Arg176Cys or Arg158Cys according to the old nomenclature) and rs429358 (NP_000032.1:Cys130Arg or Cys112Arg). Homozygotes for the ε2 allele (T/T genotype), in the absence of changes in rs429358 (T/T genotype), will form the ε2ε2 haplotype [2,3,4,5]. In more than 90% of cases, the ε2ε2 haplotype predisposes to the development of an autosomal recessive form of FD [6]. However, for the development of FD, the ε2ε2 haplotype alone is not sufficient. Additional factors leading to increased synthesis of very-low-density lipoproteins or impaired excretion of lipoprotein remnants, including impaired functioning of the heparan sulfate proteoglycan receptor, are necessary. These factors include overweight or obesity [3,7,8], insulin resistance [3,7,9], diabetes mellitus [3,10], hypothyroidism, some medications, menopause [3], and pregnancy [11,12]. In the presence of these conditions, inhibition or even degradation of the heparan sulfate proteoglycan receptor can occur, leading to the accumulation of remnants and the development of FD [3].

It was also shown that around 10% of patients could have the autosomal dominant FD associated with a single copy of the defective *APOE* allele. Approximately 30 *APOE* variants associated with the autosomal dominant FD have been reported [13,14,15]. However, there are insufficient data regarding the relationship between the described *APOE* variants and FD [15].

Previous epidemiological studies demonstrated a varied prevalence of FD (0.2–2.7%) [16,17]. Thus, it was shown that the prevalence of FD was 0.7% (one in 143) in the population of Utah, USA, and 2.7% (one in 37) among patients with premature coronary artery disease [16]. In a more recent study, the prevalence of FD among US adults in two population-based samples was 0.2–0.8% [17]. This estimate is based on analyzing plasma lipoproteins by ultracentrifugation. Recent data suggest that the real prevalence of FD may be underestimated [18,19]. The prevalence of FD in Russia has not been assessed.

In recent years, several biochemical algorithms have been developed to diagnose FD using available laboratory tests. These algorithms rely on assessing the ratios between the levels of apolipoprotein B (apoB), total cholesterol (TC), triglycerides (TG) (the apoB algorithm of Sniderman 2010 [20]), and non-high-density lipoprotein cholesterol (non-HDL-C) [21,22]. None of these criteria have been applied in Russia.

The aim of this study was to investigate the practical applicability of the diagnostic criteria for FD and the prevalence of FD in one of the European regions of Russia based on a population sample. We also analyzed the presence and pathogenicity of *APOE* variants associated with autosomal dominant FD in a large research sample.

## 2. Results

### 2.1. Applicability of Biochemical FD Criteria

We analyzed the genetic data in the population sample (*n* = 1652; Section 4.1 and Section 4.2 Materials and Methods) and identified 14 carriers of the ε2ε2 haplotype (Table 1).

In total, 50% of the carriers were men, and the median age was 50 years [46; 55]. More than half (57.1%) were obese. Coronary heart disease was present in 7.1%. Additionally, 58.3% had carotid atherosclerosis. Among subjects with the ε2ε2 haplotype, 33.3% had femoral atherosclerosis. The TG level was 2.11 [1.54; 2.60] mmol/L.

To estimate the prevalence of FD, it was necessary to establish criteria for the identification of subjects with not only the ε2ε2 haplotype but also FD. However, the diagnostic value of the biochemical algorithms for FD in subjects living in Russia has not been previously assessed.

The sensitivity and specificity of biochemical FD criteria were analyzed (for subjects without lipid-lowering therapy), as shown in Table 2.

The apoB algorithm analysis showed the low specificity of apoB level (12.0%) and TG/apoB-ratio (0.1%) for identifying subjects without the ε2ε2 haplotype. TG level ≥ 1.5 mmol/L had sufficient sensitivity (76.9%) and specificity (65.1%) to identify subjects with the ε2ε2 haplotype, as well as the non-HDL-C/apoB ratio >4.91 mmol/g. At the same time, the non-HDL-C/apoB ratio ≥3.69 mmol/g had a low specificity (1.2%) for identifying subjects without the ε2ε2 haplotype in the study population.

The apoB algorithm of Sniderman 2010 [20] is the most widely used biochemical diagnostic algorithm for FD. In order to diagnose FD according to the apoB algorithm, laboratory data must meet all the following cutoffs: apoB < 1.2 g/L, TG ≥ 1.5 mmol/L, TG/apoB < 10.0 mmol/g, and TC/apoB ≥ 6.2 mmol/g [20]. The apoB threshold represents the 75th percentile of the distribution of apoB in persons ≥20 years of age from the National Health and Nutrition Examination Survey III (NHANES III) study [23]. The apoB level is the main criterion, not only in the Sniderman algorithm, but it is also employed in other criteria, including the non-HDL-C/apoB ratio.

We studied the distribution of apoB levels in the ESSE-Ivanovo sample (for subjects without lipid-lowering therapy) (Table 3). The results showed that the level of apoB was lower (1.06 g/L (75th percentile)) compared to the NHANES III study [23].

The lower levels of apoB in the ESSE-Ivanovo sample may contribute to the low specificity of the apoB algorithm and the overestimation of the non-HDL-C/apoB ratio in subjects from the ESSE-Ivanovo. As a result, we may observe overestimated sensitivity and specificity of the non-HDL-C/apoB ratios for the cutoff >4.91 mmol/g, and the pronounced differences in sensitivity and specificity between Paquette 2020 [21] and Boot 2019 [22] criteria.

The non-HDL-C/apoB ratio in the ESSE-Ivanovo sample initially exceeded the cutoff ≥3.69 mmol/g in 98.7% of subjects. Moreover, the subjects with FD from the Boot 2019 study [22] showed a similar cutoff (the means were 4.0 mmol/g) to that of all subjects in the ESSE-Ivanovo sample, regardless of their FD status (the median was 4.51 [4.27; 4.73] mmol/g) (Table 4).

Thus, considering the results, it is recommended that a correction is made for the previously developed diagnostic FD algorithms, taking into account the characteristics of the distribution of apoB levels in the population. At the same time, the most appropriate combination for identifying subjects with FD in the current study appears to be the ε2ε2 haplotype and the TG cutoff ≥ 1.5 mmol/L.

### 2.2. The Prevalence of FD

The most common APOE haplotype in the ESSE-Ivanovo sample was ε3ε3 (63.3%). This was followed by ε3ε4 (19.0%), ε2ε3 (13.4%), ε2ε4 (2.0%), and ε4ε4 (1.4%). The prevalence of the ε2ε2 haplotype carriers was 0.8% (one in 118) (95% confidence interval (CI): 0.46–1.42) (Table 5).

The prevalence of FD based on ε2ε2 haplotype and various biochemical diagnostic FD criteria was assessed (Table 6).

Based on the apoB algorithm of Sniderman, the prevalence of FD in the Ivanovo region was 0.54%. When utilizing the non-HDL-C/apoB ratio, with thresholds set at ≥3.69 mmol/g or >4.91 mmol/g, the prevalence of FD was 0.85% and 0.79%, respectively.

When the TG ≥ 1.5 mmol/L cutoff was added to the ε2ε2 haplotype carriage, the prevalence of FD was 0.67% (one in 150) (95% CI: 0.33–1.19). Table 7 presents the characteristics of subjects with FD (presence of the ε2ε2 haplotype and TG ≥ 1.5 mmol/L).

The median age was 50 years [49; 58], and 45.5% were men. Among those with FD, 63.6% were obese. Coronary heart disease was present in 9.1%, and the majority (77.8%) had carotid atherosclerosis. Femoral atherosclerosis affected 33.3% of subjects with FD. The apoB level was 0.67 [0.5; 0.75] g/L. The distribution of TG levels was from a minimum value of 1.54 mmol/L to a maximum value of 6.0 mmol/L.

### 2.3. APOE Variants Associated with the Autosomal Dominant FD

FD has not only an autosomal recessive but also an autosomal dominant form. Approximately 30 *APOE* variants associated with the autosomal dominant FD have been previously reported [13,14,15]. However, the assessment of the pathogenicity of these variants has either not been previously carried out or has not been presented in full. Therefore, we analyzed the presence and pathogenicity of *APOE* variants associated with the autosomal dominant FD in a large general research sample (*n* = 3404; Section 4.2 Materials and Methods).

We identified six variants (one pathogenic, two likely pathogenic, and three variants of uncertain significance) that are associated with the autosomal dominant FD. Four of these variants have been previously described [14,15]. Additionally, we identified two rare *APOE* variants that have not been previously associated with FD. Appendix A presents characteristics of all detected *APOE* variants.

The median age of subjects with all identified variants was 48 years [40; 54], and 30.8% were men. The majority (92.3%) were at least overweight, and 15.4% of them were obese. One subject with a pathogenic (p.Glu63ArgfsTer15) *APOE* variant and another subject with a likely pathogenic (rs121918393, *APOE2* Heidelberg) *APOE* variant had xanthomas (tendon or cutaneous eruptive, respectively). The TG level among all subjects was less than 1.7 mmol/L, but subjects with pathogenic or probably pathogenic *APOE* variants had a TG level greater than 1.5 mmol/L. Table 8 presents the characteristics of subjects with available clinical data (*n* = 13).

Only one of the previously described variants, *APOE2* Heidelberg (NM_000041.4:c.460C>T; NP_000032.1:p.Arg154Cys; rs121918393), was classified as likely pathogenic according to ACMG/AMP2015. The prevalence of this variant in the population sample was 0.18% (one in 551) (95% CI: 0.04–0.53). Since this variant, *APOE2* Heidelberg, was the only one that has been directly linked with FD and the association of new variants with FD remains to be explored, autosomal dominant FD was not included in the overall estimation of FD prevalence in the current study.

## 3. Discussion

### 3.1. Applicability of Biochemical FD Criteria for Identifying Subjects with the Autosomal Recessive FD

In the majority of cases, the ε2ε2 haplotype predisposes to the development of an autosomal recessive FD [6]. Therefore, in the first stage, we analyzed the genetic data in a population sample and identified carriers of the ε2ε2 haplotype. As a result, 14 subjects from the ESSE-Ivanovo population-based cohort were found to have the ε2ε2 haplotype.

However, the presence of only one genetic basis (ε2ε2 haplotype) is not sufficient for the development of an autosomal recessive FD. Additional factors are required. These factors are highly diverse, and their contribution to the development of FD is still being studied [8]. Furthermore, genetically based hyperlipidemias may exhibit similar phenotypic features. For this reason, biochemical algorithms and criteria have been developed to help clinicians identify patients with FD.

We investigated the applicability of the most accessible biochemical algorithms for diagnosing FD in clinical practice. The diagnostic value of these biochemical algorithms for FD in subjects living in Russia has not been previously assessed. Thus, we analyzed the sensitivity and specificity of biochemical FD criteria and found a low specificity of the apoB level for identifying subjects without the ε2ε2 haplotype. This may be due to the initially lower levels of apoB in subjects from the ESSE-Ivanovo population sample (75th percentile—1.06 g/L) compared to the apoB algorithm (75th percentile—1.2 g/L according to the NHANES III study [23]). This could lead to non-carriers of the ε2ε2 haplotype being incorrectly identified as carriers.

The overestimated sensitivity and specificity of the non-HDL-C/apoB ratios for the cutoff > 4.91 mmol/g, and the differences in sensitivity and specificity between the non-HDL-C/apoB ratio cutoffs ≥ 3.69 mmol/g and >4.91 mmol/g may also be associated with a lower level of apoB in the ESSE-Ivanovo sample, as described in our study results. Additionally, the level of apoB may have interindividual variability depending on concomitant metabolic diseases and may decrease due to the intake of lipid-lowering therapy. This casts doubt on the unambiguous use of apoB level as the primary or main indicator for identifying FD in clinical practice [24,25].

On the other hand, we found that the TG cutoff ≥ 1.5 mmol/L had sufficiently high sensitivity and specificity (76.9% and 65.1%, respectively) in identifying carriers of the ε2ε2 haplotype. This finding is consistent with the Varghese 2021 study, which showed that the TG cutoff ≥ 1.5 mmol/L was optimal for identifying individuals with FD. Increasing the TG cutoff in the apoB algorithm may result in missing subjects with a less pronounced phenotype of FD [26].

Thus, algorithms that include the apoB level require correction, taking into account the characteristics of the distribution of apoB levels in the population. Additionally, the TG cutoff ≥ 1.5 mmol/L may be a useful tool in identifying subjects with FD.

### 3.2. The Prevalence of the Autosomal Recessive FD

The study detected a high prevalence of FD (0.5–0.8%, or one in 118–184) in one of the European regions of Russia, based on ε2ε2 haplotype and various simple biochemical diagnostic FD criteria. This prevalence of FD is consistent with findings from other population-based studies. For instance, the prevalence of FD was 0.7% (one in 143) in the population of Utah, USA, where FD was estimated using a ratio of very-low-density lipoprotein/TG ≥ 0.30 determined by ultracentrifugation with TG levels > 150 mg/dL [16]. In a study by Pallazola 2019, the prevalence of FD among US adults in two population-based samples was 0.2–0.8%. This estimate was also based on analyzing plasma lipoproteins by ultracentrifugation [17]. When using the apoB algorithm, the prevalence of FD was found to be 2.0%, which is higher than the prevalence observed in the current study (0.54% according to the apoB algorithm) [17].

Considering the obtained results, which indicate a low specificity of the apoB level in identifying subjects without the ε2ε2 haplotype, as well as the initially lower levels of apoB in subjects from the ESSE-Ivanovo population sample, and the need for correction of algorithms that include the apoB level, in this current study we estimated the prevalence of FD based on the ε2ε2 haplotype and TG levels ≥ 1.5 mmol/L. Thus, the prevalence of FD was 0.67%, or 1 in 150.

According to the European Medicines Agency (EMA), a disease with a maximum prevalence of 5 per 10,000 people is considered rare [27]. Therefore, FD is not a rare disease (with a prevalence of 66 per 10,000 people in the Ivanovo region). Moreover, the prevalence of FD in the Ivanovo region is comparable to that of familial hypercholesterolemia, the prevalence of which we have previously reported (1 per 111 people [28], *p* = 0.453). However, the prevalence of familial hypercholesterolemia has been widely studied in various populations, including Russian [29], and is covered at the global level [30,31]. In contrast, the equally common and no less atherogenic FD remains understudied.

Only 5.7% of subjects in the ESSE-Ivanovo sample were taking statins, and more than half of patients with coronary artery disease (58.9%) were not receiving lipid-lowering therapy. It is important to emphasize that none of the ESSE-Ivanovo subjects had previously been diagnosed with FD. The optimal solution for patients with FD is combined lipid-lowering therapy, which includes fenofibrate [30]. However, only one patient with FD received statins, and none of the patients received combined lipid-lowering therapy. Additionally, none of the patients with FD received fenofibrate or omega-3 polyunsaturated fatty acids, which are available in Russia. Therefore, the common and highly atherogenic FD remains underdiagnosed, while the use of irrational lipid-lowering therapy results in undertreatment of patients with FD.

### 3.3. APOE Variants Associated with the Autosomal Dominant FD

FD has not only an autosomal recessive but also an autosomal dominant form. We have identified four *APOE* variants previously associated with the autosomal dominant FD [14,15]. However, only one of these variants, *APOE2* Heidelberg (NM_000041.4:c.460C>T; NP_000032.1:p.Arg154Cys; rs121918393), was classified as likely pathogenic according to ACMG/AMP2015 pathogenicity criteria (PM1 [32], PM2, PP1_Moderate [32], PP3, PP4, and PP5). The pathogenicity of the three other previously described *APOE* variants (rs267606664, rs199768005, and rs267606661) in the development of the autosomal dominant FD remains uncertain. These variants have a prevalence exceeding 0.01% in control samples, and there are limited data on their segregation with the FD phenotype.

In addition, we have identified two new frameshift *APOE* variants that are pathogenic (NM_000041.4:c.184_187del, NP_000032.1: p.Glu63ArgfsTer15) or likely pathogenic (NM_000041.4: c.434_461del, NP_000032.1: p.Gly145AlafsTer97) in the development of the autosomal dominant FD. It would be important to analyze their segregation in the proband’s relatives. Notably, according to the findings of this study, subjects with pathogenic or likely pathogenic variants associated with the autosomal dominant FD were the only ones with xanthomas, alongside TG levels ≥ 1.5 mmol/L.

In the current study we could not include the autosomal dominant FD in the overall estimation of FD prevalence. Further studies analyzing the phenotype–genotype relationship and conducting functional analysis can contribute to establishing the causal role of *APOE* variants in the development of the autosomal dominant FD. Determining the contribution of these variants to the development of the autosomal dominant FD will help increase the detection of individuals with this disease.

## 4. Materials and Methods

### 4.1. Sampling

The population sample was selected from the “Epidemiology of Cardiovascular Diseases and Risk Factors in Regions of the Russian Federation” (ESSE-RF) study which was a cross-sectional study conducted from 2012 to 2013 across 13 regions of Russia [33]. The current study included a sample from the Ivanovo region (ESSE-Ivanovo) [34] (median age was 49 years old [39; 57]; 37.2% were men, *n* = 1652; Appendix A). The ESSE-Ivanovo sample (*n* = 1652) was used to identify the ε2ε2 haplotype carriers and investigate the prevalence of FD. From this sample, we selected 1550 subjects without lipid-lowering therapy to assess the sensitivity and specificity of biochemical FD criteria and the distribution of apoB levels.

The ESSE-FH-RF sample consisted of participants from the ESSE-FH-RF study who had a clinical diagnosis of definite or probable heterozygous FH (*n* = 161) [28]. 

The Russian patient sample (RPS) was formed of patients with diverse medical conditions who were observed at the National Medical Research Center (NMRC) for Therapy and Preventive Medicine (Moscow, Russia) (*n* = 1591).

### 4.2. Applicability of Biochemical FD Criteria

The lipid concentrations (apoB, LDL-C, HDL-C, TC and TG) were measured with the Abbott Architect C-8000 system (Abbott Laboratories, North Chicago, IL, USA). The HDL-C, LDL-C, TG, TC, or apoB levels were available for all 1652 subjects from the ESSE-Ivanovo sample.

We first analyzed genetic data in a population sample (ESSE-Ivanovo, *n* = 1652) to identify the ε2ε2 haplotype carriers. Then, we assessed the sensitivity and specificity of biochemical FD criteria, such as the apoB algorithm [20] (combination of criteria apoB < 1.2 g/L, TG ≥ 1.5 mmol/L, TG/apoB < 10.0 mmol/g, TC/apoB ≥ 6.2 mmol/g), the non-HDL-C/apoB ratio ≥ 3.69 mmol/g [21], and the non-HDL-C/apoB ratio > 4.91 mmol/g [22] (ESSE-Ivanovo population sample, subjects without lipid-lowering therapy, *n* = 1550) (Figure 1A). The ε2ε2 haplotype carriers were considered conditionally ill, whereas conditionally healthy are those without the ε2ε2 haplotype. The sensitivity was calculated as the ratio of true positive (the number of cases correctly identified as ε2ε2 haplotype carriers)/true positive + false negative (the number of cases incorrectly identified as healthy). The specificity was calculated as the ratio of true negative (the number of cases correctly identified as healthy)/true negative + false positive (the number of cases incorrectly identified as ε2ε2 haplotype carriers) [35]. We also analyzed the distribution of apoB levels (ESSE-Ivanovo population sample, subjects without lipid-lowering therapy, *n* = 1550).

The prevalence of FD was determined based on the ESSE-Ivanovo population sample (*n* = 1652) (Figure 1A).

Genetic data from the total of 3404 samples (ESSE-Ivanovo, ESSE-FH-RF and RPS) were used to identify previously reported and new *APOE* variants associated with the autosomal dominant FD (Figure 1B).

### 4.3. Next-Generation Sequencing

Genomic DNA was isolated from peripheral blood with the QIAamp DNA Blood Mini Kit (Qiagen, Hilden, Germany). DNA concentration was measured on the Qubit 4.0 fluorimeter (Thermo Fisher Scientific, Waltham, MA, USA). For next-generation sequencing (NGS), the libraries were prepared using either the SeqCap EZ Prime Choice Library kit (Roche, Basel, Switzerland) for targeted sequencing (custom panel) or the IDT-Illumina TruSeq DNA Exome protocol (Illumina, San Diego, CA, USA) for exome libraries. The custom panel consisted of 25 genes (CDS + 25 bp padding), associated with lipid metabolism disorders, including APOE (ABCA1, ABCG5, ABCG8, ANGPTL3, APOA1, APOA5, APOB, APOC2, APOC3, APOE, CETP, GPD1, GPIHBP1, LCAT, LDLR, LDLRAP1, LIPC, LIPI, LMF1, LPL, MTTP, PCSK9, SAR1B, STAP1, USF1). Sequencing was performed on a Nextseq 550 (Illumina, San Diego, CA, USA). All stages of sequencing were conducted according to the manufacturers’ protocols.

### 4.4. Bioinformatic Processing of Sequencing Data and Clinical Interpretation

All bioinformatic analyses were described in more detail in a previous study of the ESSE-Ivanovo sample [34].

The processing of sequencing data and quality control evaluation were performed using the custom-designed pipeline [34] based on GATK 3.8 [36]. Paired-end reads were aligned to the GRCh37/hg19 reference genome. As a result, individual VCF files were obtained for each subject, containing a list of variants, their genomic coordinates, coverage data, and other characteristics. Low-quality variants, which are presumably sequencing errors, were filtered out. The coverage depth of the reference and alternative alleles, the quality of reads and mapping, and other relevant factors were reported and analyzed.

The ε2ε2 haplotype of the *APOE* gene was determined to be homozygous for the ε2 allele (rs7412) in combination with no alteration in rs429358 (Table 9).

Previously reported *APOE* variants associated with the autosomal dominant FD, as well as rare *APOE* variants, were analyzed [14,15].

The annotation of rare *APOE* variants was performed with OMIM [37], gnomAD (v2.1.1) [38]), ClinVar (2021/01/10) [39], Human Gene Mutation Database (HGMD) [40], Leiden Open Variation Database (LOVD) [41], dbSNP [42] databases, and data from the literature. The clinical interpretation is based on the American College of Medical Genetics and Genomics/Association for Molecular Pathology (ACMG/AMP2015) guidelines [43].

### 4.5. Sanger Sequencing

The validation of results by Sanger sequencing was conducted for the c.526C>T (p.Arg176Cys) variant, all known variants associated with the autosomal dominant form of FD, as well as rare *APOE* variants. The DNA sequencer Applied Biosystem 3500 Genetic Analyzer (Thermo Fisher Scientific, Waltham, MA, USA) was used for Sanger sequencing together with the ABI PRISM BigDye Terminator v3.1 reagent kit (Thermo Fisher Scientific, Waltham, MA, USA), following the manufacturer’s protocol.

### 4.6. Statistical Analyses

Statistical analyses were performed using R v. 4.1.2 (R Foundation for Statistical Computing, Vienna, Austria) [44]. The data are presented as a median (25th–75th percentile). We calculated the prevalence of FD by dividing the number of people with FD by the total sample size. The prevalence of FD was calculated as a percentage for all participants. The Clopper–Pearson exact method was used for the estimation of the 95% confidence interval. A *p*-value of less than 0.05 was considered statistically significant. 

### 4.7. Limitations of the Study

The prevalence of FD, as well as the distribution of apoB levels and the sensitivity and specificity of biochemical FD criteria, were estimated in the Ivanovo region, which is just one region in Russia. While we cannot extrapolate our results to the entire Russian population, the Ivanovo region belongs to the European region of Russia and is representative of similar regions. The ethnic composition of the Ivanovo region is 95.57% Russian [45]. Therefore, the data obtained from our study can be applied to similar regions in Russia.

The sensitivity and specificity of biochemical FD criteria in the ESSE-Ivanovo sample were evaluated relative to the ε2ε2 haplotype carriers. We assessed the applicability of previously developed biochemical FD criteria available for clinical practice. The lipoprotein ultracentrifugation or electrophoresis were not used in this study.

## 5. Conclusions

A high prevalence of FD (one in 150) was detected in one of the European regions of Russia based on the ε2ε2 haplotype and TG levels ≥ 1.5 mmol/L. Underdiagnosis of FD is typical for Russia. It is not advisable to use previously developed diagnostic FD algorithms, including the apoB level, in Russia. These algorithms need to be adjusted for the population apoB levels. At the same time, the TG cutoff ≥ 1.5 mmol/L may be a useful tool in identifying subjects with FD.

## Figures and Tables

**Figure 1 ijms-24-13159-f001:**
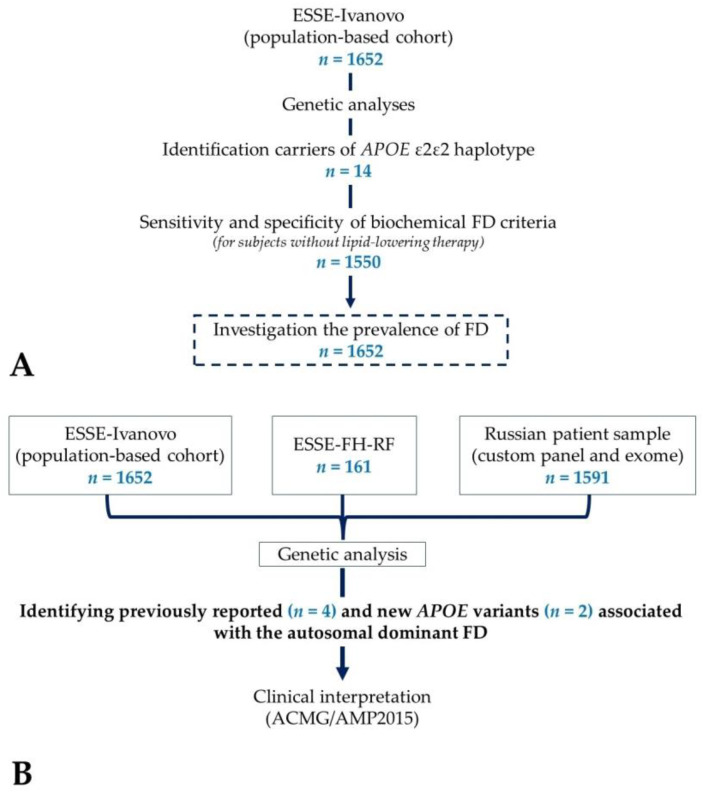
The study design: (**A**) Identification of carriers of the *APOE* ε2ε2 haplotype, assessment of the sensitivity and specificity of biochemical FD criteria, and prevalence of FD; (**B**) Identification of previously reported and new *APOE* variants associated with the autosomal dominant FD, analysis of the genotype–phenotype relationship.

**Table 1 ijms-24-13159-t001:** Characteristics of subjects with the ε2ε2 haplotype.

Parameter	The Subjects with the ε2ε2 Haplotype (*n* = 14)
Men, *n* (%)	7 (50.0%)
Age, years	50 [46; 55]
BMI, kg/m^2^	30.5 [26.7; 32.9]
Presence of diabetes, *n* (%)	0
Presence of arterial hypertension, *n* (%)	12 (85.7%)
CHD, *n* (%)	1 (7.1%)
Carotid atherosclerosis, *n* (%)	7 (58.3%) *n* = 12
Femoral atherosclerosis, *n* (%)	4 (33.3%) *n* = 12
Statins, *n* (%)	1 (7.1%)
apoB, g/L	0.64 [0.52; 0.73]
TC, mmol/L	5.2 [4.29; 5.83]
TG, mmol/L	2.11 [1.54; 2.60]
LDL-C, mmol/L	2.14 [1.78; 2.44]
HDL-C, mmol/L	1.18 [1.10; 1.54]

apoB—apolipoprotein B; BMI—body mass index; CHD—coronary heart disease; HDL-C—high-density lipoprotein cholesterol; LDL-C—low-density lipoprotein cholesterol; TG—triglycerides; TC—total cholesterol.

**Table 2 ijms-24-13159-t002:** The sensitivity and specificity of biochemical FD criteria in the ESSE-Ivanovo sample (*n* = 1550).

Criterion	Passed the Criterion for Subjects with the ε2ε2 Haplotype (*n* = 13) ^1^	Did Not Pass the Criterion for Subjects without the ε2ε2 Haplotype (*n* = 1537)	Sensitivity, %	Specificity, %
apoB algorithm of Sniderman 2010 [20]				
apoB < 1.2 g/L	13	185	100	12.0
TG ≥ 1.5 mmol/L	10	1001	76.9	65.1
TG/apoB < 10.0 mmol/g	13	2	100	0.1
TC/apoB ≥ 6.2 mmol/g	12	842	92.3	54.8
non-HDL-C/apoB ratio				
non-HDL-C/apoB ≥ 3.69 mmol/g [21]	13	19	100	1.2
non-HDL-C/apoB > 4.91 mmol/g [22]	12	1364	92.3	88.7

^1^ One subject, who was taking statins, was excluded from this analysis. As a result, the total number of subjects included in the results is 13 instead of 14. apoB—apolipoprotein B; non-HDL-C—non-high-density lipoprotein cholesterol; TG—triglycerides.

**Table 3 ijms-24-13159-t003:** The distribution of apoB levels in the ESSE-Ivanovo sample (*n* = 1550).

Age, Years	Number of Subjects	apoB, g/L		
		50th Percentile	75th Percentile	95th Percentile
All subjects				
25–64	1550	0.89	1.06	1.35
25–34	285	0.73	0.86	1.14
35–44	325	0.84	0.98	1.24
45–54	472	0.93	1.08	1.37
55–64	468	0.98	1.16	1.43
Men				
25–64	574	0.88	1.05	1.33
25–34	167	0.78	0.92	1.18
35–44	142	0.87	1.02	1.29
45–54	151	0.96	1.11	1.43
55–64	114	0.96	1.13	1.36
Women				
25–64	976	0.90	1.07	1.35
25–34	118	0.67	0.79	1.06
35–44	183	0.81	0.95	1.14
45–54	321	0.92	1.06	1.34
55–64	354	1.00	1.17	1.45

apoB—apolipoprotein B.

**Table 4 ijms-24-13159-t004:** The distribution of apoB levels and the non-HDL-C/apoB ratio in the ESSE-Ivanovo sample and other studies.

Parameter	ESSE-Ivanovo Sample	Non-HDL-C/apoB Ratio ≥ 3.69 Mmol/g, Paquette 2020 [21]		Non-HDL-C/apoB Ratio > 4.91 mmol/g, Boot 2019 [22]	
	All Subjects (*n* = 1550)	with FD Subjects (*n* = 188)	without FD Subjects (*n* = 4703)	with FD Subjects (*n* = 63)	without FD Subjects (*n* = 1397)
apoB, g/L	0.89 [0.74; 1.06] ^1^	1.30 [1.04–1.83] ^1^	1.81 [1.50–2.17] ^1^	1.35 (0.4) ^2^	1.00 (0.3) ^2^
non-HDL-C/apoB, mmol/g	4.51 [4.27; 4.73] ^1^	5.36 [4.54–6.64] ^1^	3.38 [2.97–3.87] ^1^	4.0 (0.5) ^2^	7.3 (1.5) ^2^

^1^ The data are presented as median (25th–75th percentile). ^2^ The means with standard deviations in parentheses (Boot 2019 [22]). apoB—apolipoprotein B; FD—familial dysbetalipoproteinemia; non-HDL-C—non-high-density lipoprotein cholesterol.

**Table 5 ijms-24-13159-t005:** Characteristics of the ESSE-Ivanovo sample across *APOE* haplotypes.

ESSE-Ivanovo Sample, *n*	*APOE* Haplotype, *n* (%) (95% Confidence Interval)					
	ε3ε3	ε2ε2	ε4ε4	ε2ε3	ε3ε4	ε2ε4
1652	1046 (63.3) (60.94–65.65)	14 (0.8) (0.46–1.42)	23 (1.4) (0.88–2.08)	222 (13.4) (11.83–15.18)	314 (19.0) (17.14–20.98)	33 (2.0) (1.38–2.79)

**Table 6 ijms-24-13159-t006:** The prevalence of FD based on ε2ε2 haplotype and various biochemical diagnostic FD criteria.

Criterion/Algorithm	Number of Subjects with FD, *n*	FD Prevalence, %	95% Confidence Interval
ε2ε2 and apoB algorithm of Sniderman 2010 [20]	9	0.54 (1 in 184)	0.25–1.03
ε2ε2 and non-HDL-C/apoB ≥3.69 mmol/g [21]	14	0.85 (1 in 118)	0.46–1.42
ε2ε2 and non-HDL-C/apoB >4.91 mmol/g [22]	13	0.79 (1 in 127)	0.42–1.34
ε2ε2 and TG ≥ 1.5 mmol/L	11	0.67 (1 in 150)	0.33–1.19

apoB—apolipoprotein B; FD—familial dysbetalipoproteinemia; non-HDL-C—non-high-density lipoprotein cholesterol; TG—triglycerides.

**Table 7 ijms-24-13159-t007:** Characteristics of subjects with FD based on the ε2ε2 haplotype and TG levels ≥ 1.5 mmol/L.

Parameter	The Subjects with FD (*n* = 11)
Men, *n* (%)	5 (45.5%)
Age, years	50 [49; 58]
BMI, kg/m^2^	31.6 [29.0; 33.9]
Presence of diabetes, *n* (%)	0
Presence of arterial hypertension, *n* (%)	10 (90.9%)
CHD, *n* (%)	1 (9.1%)
Carotid atherosclerosis, *n* (%)	7 (77.8%) *n* = 9
Femoral atherosclerosis, *n* (%)	3 (33.3%) *n* = 9
Statins, *n* (%)	1 (9.1%) ^1^
apoB, g/L	0.67 [0.54; 0.75]
TC, mmol/L	5.4 [4.47; 6.28]
TG, mmol/L	2.31 [1.86; 2.81]
LDL-C, mmol/L	2.12 [1.92; 2.48]
HDL-C, mmol/L	1.19 [1.05; 1.51]

^1^ This patient with FD, who was taking statins, was excluded from the analysis of the distribution of apoB levels, the sensitivity and specificity of biochemical FD criteria. apoB—apolipoprotein B; BMI—body mass index; CHD—coronary heart disease; HDL-C—high-density lipoprotein cholesterol; LDL-C—low-density lipoprotein cholesterol; TG—triglycerides; TC—total cholesterol.

**Table 8 ijms-24-13159-t008:** Characteristics of subjects with variants associated with the autosomal dominant FD.

Parameter	All Subjects (*n* = 13)	Subjects with VUS (*n* = 7)	Subjects with LP or P Variants (*n* = 6)
Men, *n* (%)	4 (30.8%)	2 (28.6%)	2 (33.3%)
Age, years	48 [40; 54]	48 [30; 54]	45 [40; 57]
BMI, kg/m^2^	26.3 [26.0; 28.9]	26.3 [26.0; 28.9]	27.4 [25.7; 30.0]
Presence of xanthoma, *n* (%)	2 (15.4%)	0	2 (33.3%)
CHD, *n* (%)	0	0	0
TG, mmol/L	1.36 [1.11; 1.95]	1.36 [1.11; 1.70]	1.55 [0.69; 4.75]
LDL-C, mmol/L	3.0 [2.32; 3.85]	3.04 [1.66; 3.94]	2.96 [2.32; 3.54]
HDL-C, mmol/L	1.23 [1.0; 1.81]	1.18 [0.98; 2.07]	1.27 [1.09; 1.66]

BMI—body mass index; CHD—coronary heart disease; HDL-C—high-density lipoprotein cholesterol; LDL-C—low-density lipoprotein cholesterol; LP—likely pathogenic; P—pathogenic; TG—triglycerides; VUS—variant of uncertain significance.

**Table 9 ijms-24-13159-t009:** *APOE* variants that define the haplotypes.

Gene	Variant	Genomic Coordinates (GRCh37)	Haplotype						HGVSc (NM_000041.4), HGVSp (NP_000032.1)
			**ε3ε3**	**ε2ε2**	**ε4ε4**	**ε2ε3**	**ε3ε4**	**ε2ε4**	
*APOE*	rs7412	chr19:45412079_C/T	C/C	T/T	C/C	C/T	C/C	C/T	c.526C>T, p.Arg176Cys
*APOE*	rs429358	chr19:45411941_T/C	T/T	T/T	C/C	T/T	T/C	T/C	c.388T>C, p.Cys130Arg

HGVSc—Human Genome Variation Society coding sequence name; HGVSp—Human Genome Variation Society protein sequence name.

## Data Availability

The data used and/or analyzed during the current study are available from the corresponding authors on reasonable request. Individual genotype information cannot be made available in order to protect participant privacy.

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
