# Peer review of "Applicability of Diagnostic Criteria and High Prevalence of Familial Dysbetalipoproteinemia in Russia: A Pilot Study"

_ijms, 2023, doi:10.3390/ijms241713159_

Round 1
Reviewer 1 Report
The manuscript, entitled "Applicability of Diagnostic Criteria and Prevalence of Familial Dysbetalipoproteinemia in Russia" shows prevalence of FD in a strange manner in the Russian Federation. The title of the article does not correspond to reality, since the authors studied only one region in the Russian Federation, the ethnicity of which is not clear. In addition, probably (it is not clear from the article) the prevalence estimates included moderated samples according to unknown inclusion criteria.
Second aim of this study was to assess the usefulness of a diagnostic criteria for FD in the RF, specifically biochemical markers. The result on the applicability of the diagnostic criteria seems to remain unanswered.
The authors do not offer any hypotheses for the study.
Author argues that there is not enough data on Sniderman's (2010) FD diagnostic criteria algorithm in actual clinical application. Indeed, these diagnostic criteria have been developed in relatively small cohorts and often using inappropriate criteria for the diagnosis of FD and this issue is still widely discussed. In this respect, there is a weak novelty to the study. From this point of view, the usefulness of this study would not be to criticize a long-known fact, but to develop its own approach or new criteria more appropriate for the FD or at least for FD in the RF.
Indeed, in addition to genetic conditions, nutritional conditions can also modify biochemical indicators. The Russian Federation is one of the most multinational countries with a variety of cuisines. In connection with the above, the authors should have at least filtered the prevalence of FD by racial/ethnic, which was not done.
The manuscript lacks the limitations of interpreting the results. For example, the results for only one region of the Russian Federation cannot be judged for the entire country, since living conditions vary dramatically from region to region.
There are big questions about the design of the FD prevalence estimation study. The text of the manuscript does not match the picture of the study design. For example, the text describes that the Ivanovo population was used to estimate FD prevalence, while the picture shows that all samples were used for this purpose. The same questions to the whole picture of the research design, almost complete inconsistency with what is described in the text.
The authors used only 54% of recent publications (within the last 5 years). The literature sources do not contain excessive self-citation.
The methods are written quite well, but there is some redundant information in places (f.e. 232, 237). The number of people in the sample varies in methods (1,656), results (1,550), and supplementary data (1,652 with biochemical measures). The manuscript does not contain sufficient information on what criteria the specimens were chosen for the study. The picture shows "custom panel" that it is not described in the methods.
In 106-108 is not a very clear decision of the authors.
It is not quite clear why the sample was expanded, if all patients were found in the Ivanovo region, according to Table 3? Thus, the other data had no effect on the result of the study, and therefore can be excluded without loss of value.
Figure 2 is presented unnecessarily.
In the conclusion, the authors summarize the results, but do not draw any conclusions, except that the apoB algorithm needs to be corrected. The statement of the findings, as made in the Conclusion, does not improve the overall impression of the manuscript. The usefulness of the results for the scientific community, could be in a comparative aspect with other neighboring regions, or countries, which also was not done.
Reviewer 2 Report
The aim of the reviewed article was the assessment of applicability of some diagnostic algorithms for familial dysbetalipoproteinemia. The authors showed that diagnostic algorithms including apoB levels require correction, and the triglycerides cutoff ≥1.5mmol/L may be a useful tool in identifying subjects with familial dysbetalipoproteinemia.
In my opinion, the manuscript is clear, relevant to the field, and presented in a well-structured manner.
The cited references are not predominantly recent publications (2018-2022), but it can be explained by the fact that a lot of fundamental data on the topic of familial dysbetalipoproteinemia need to be taken into account.
I have no complaints about the research methods, the figures and tables. The authors conducted a biochemical, genetic, bioinformatics study, as well as a comparison with the literature data to identify the most appropriate algorithm for familial dysbetalipoproteinemia diagnostics. The conclusions of the article are also consistent with the results of the study. The ethics statements and data availability statements are adequate.
Thus, I do not see methodological errors and consider the article worth publishing
Author Response
We would like to express our gratitude to Reviewer 2 for their time and constructive comments on our manuscript.
There have not been as many studies dedicated to familial dysbetalipoproteinemia in the past five years, especially when compared to other hyperlipidemias. This emphasizes the significance of our study and the presentation of its findings. We used both foundational research and recent studies to comprehensively highlight the issue.
We sincerely thank the Reviewer 2 for their comments.
Reviewer 3 Report
This study can suggested a good information about the presence and pathogenicity of APOE variants associated with the autosomal dominant FD. Howeer ii is need to revise on the focus of next contents.
Title is a different from the sentence of purpose in introduction. So it is a need to revise the title.
The number of subjects is insufficient state for tha analysis of APOE variants의 haplotype.
I wish I could explain the results in an easier to understand way.
Please explain more systematic discussions based on the obtained results.
It is hoped that the conclusions will be expressed more systematically, focusing on the original research purpose.
Minor editing of English language is required.
Round 2
Reviewer 1 Report
Thank you to the authors for your thoughtful approach in responding to the reviewer's comments. Indeed, the authors did a good job of responding to the reviewer's comments. However, readers will not see any of this, so I would like to see the author's vision in the article rather than in the response to the reviewer, as similar questions may be left unanswered by readers, which will reduce the overall value of this work.
The manuscript, under the modified title "Applicability of Diagnostic Criteria and Prevalence of Familial Dysbetalipoproteinemia in One of the European Regions of Russia" reveals some features of the prevalence and characteristics of FD in the Russian Federation. The phrase "One of the European regions of Russia" in the title reflects the reality of the article itself, but almost completely negates its scientific value. The point is that an article with this wording would set a precedent that would encourage such flawed articles to appear as full-fledged and thereby reduce their overall value to the scientific community and the journal.
Therefore, in my opinion, the title should be drastically changed to "Pilot study of the prevalence of FD in the Russian Federation", for example. There is no need to reflect in the title the applicability or correction of diagnostic criteria, as it is only one of the findings.
In any case, the original title carries no information about the applicability or inapplicability of the diagnostic criteria of FD to the Russian Federation, nor about its prevalence. The article could have been titled at least this way: “High prevalence of FD in the Russian Federation: a pilot study” or something like that. Which may indeed be of interest to the reader, for the details of which he or she can consult the article.
Authors should clearly present the research design. In its current form, the study design (Figure 2) is still very vague. The coherence of the study design is weak because of the presence of three independent objectives, the unifying factor of which is FD. Even so, some aspects of the study design remain questionable.
Assessment of diagnostic criteria. The evaluation of "applicability" was conducted in two stages, not as shown in Figure 2A. In the first stage, diagnostic criteria were evaluated using biochemical tests and genotyping of autosomal recessive alleles, with autosomal dominant alleles excluded because of the lack of data on their clinical effect.
The part of the study devoted to the search for autosomal dominant alleles does not fit into the overall design and is a separate study that does not affect the overall result of the study. However, it was included by the authors in the study due to the discovery of new previously unidentified alleles. Consequently, this part of the work can be moved to the end of the results rather than to the beginning.
In addition, the design of the study can be misleading from Figure 2A. Only the Ivanovo region was used to assess both the prevalence of FD and the applicability of diagnostic criteria for FD. The other samples are part of the search for autosomal dominant FD alleles and do not participate in the overall project. This part of the work can be completely reduced to those patients in whom interesting alleles are found. As it stands, the study design disturbs the overall perception of the work and, if this part is reduced, it will have no effect on the prevalence estimate of FD, the applicability of the diagnostic criteria for FD, or the conclusions in general.
Second step of the study. Line 128 is unclear; it is not clear from the sentence whether "n=33" refers to lipid spectrum genes or to patients filtered using such a method. Figure 2A shows that 1591 patients underwent NGS, and for the Ivanovo population and another 161 patients, it is unclear which method was used. Most likely NGS, but it is not clear which panel. In addition, it is not clear what is meant by a custom panel. Only the reagents for preparing the custom target for sequencing are given, without specifying how many and which genes were taken for sequencing. The description of the NGS method is still very sparse. In any case, the second step of the study was biochemical testing and determination of the ε2ε2 genotype to assess specificity and sensitivity of …, more about that later.
Here, conclusions were drawn, the most appropriate criteria were selected, their levels were adjusted if necessary. This may have been an obvious design, but in reality the authors simply ignored their own results and decided to take one of the existing criteria and use it as is as the most applicable. Not only that, the authors chose to take a diagnostic criterion (TG ≥1.5 mmol/L) with a sensitivity of 77% and specificity of 65% instead of a diagnostic criterion (Boot 2019) giving 92% and 89%, sensitivity and specificity, respectively. The authors' decision raises great questions.
Table 4. Add in a note what is indicated in parentheses, standard error, standard deviation, or something else.
Table 4 clearly indicates that when the diagnostic criterion with a TG cutoff ≥1.5 mmol/L with the ε2ε2 genotype was applied, 10 individuals were identified, while in line 176 had already become 11. From this, the authors infer a prevalence of FD (1:151). Whether it is true or not is no longer clear.
However, at the beginning of their results, the authors conclude that only 22 of 37 ε2ε2 genotype carriers passed the apoB-algorithm, with patients with FD excluded. The authors thereby concluded that the apoB-algorithm is not suitable. It is not clear where the rest of the patients from the beginning of the study went, i.e., only 13 carriers out of 37 were subsequently used. The answer might be that line 140 mentions that apoB-algorithm patients were excluded, but then there should have been 11, not 13 according to line 131. The authors do not describe the sequence of their actions and there is confusion in the data.
Most likely, the authors used the entire sample (n=3408) and found E2E2 carriers and applied the apoB algorithm to them, but then simply excluded them from further results and discussion. Why it was necessary to do it is not quite clear. It is possible to exclude this part completely and leave only the Ivanov sample as it is and nothing will change. On the other hand, why then only the Ivanovo population was taken to assess the applicability of TG, since TG levels were available for the entire sample. This part of the research is very confusing.
A big question remains why the specificity and sensitivity of the diagnostic criteria were evaluated relative to the ε2ε2 genotype. Consequently, the diagnostic criteria determine the ε2ε2 genotype, not FD. The usefulness of this technique remains questionable.
Overall, the article leaves even more questions about the scientific soundness of the study.
Reviewer 3 Report
It is considered well-modified according to the review results pointed out. Thank you for your effort.
Minor editing English language required.
Round 3
Reviewer 1 Report
Thanks to the authors for taking into account most of the reviewer's comments and revising their vision of the article, which improved its overall impression. However, there are still some controversial points.
Major:
Line 27. “However, there is not enough data on their use in real-world clinical implementation” is a controversial statement for a number of reasons, first of all, what does it mean according to the authors to have enough data and enough for what? On the other hand, the authors note that many countries use FD diagnostic criteria and even provide data, but it turns out that for the authors it is not enough. This comment has been made before, but has been ignored by the authors.
It's not quite clear what the authors mean by "ultracentrifugation criteria". There is no such term.
The authors probably meant Vertical Auto Profile ultracentrifugation lipid profile analysis. The authors deliberately took the term out of the context of the original article and used the term in a distorted or meaningless way. Indeed, the gold standard for FD diagnosis was a ratio of VLDL-cholesterol to VLDL-triglycerides > 0.30, as determined by ultracentrifugation, but this method is no longer used in clinical chemistry laboratories (Blum and Type, 2016).
There is a lack of information about other genotypes mentioned by the authors in the introduction. Indeed, on the one hand, it is not obligatory, but it gives a general idea of the prevalence of APOE alleles in Russia and could fit 1-2 sentences at the very beginning of the results (section 2.2) as general descriptive statistics of the genetic component.
The authors state that the apoB-based diagnostic criteria did not identify FD in FD patients (Line 108), arguing that they had xanthomas that are not actually pathognomonic for FD. In general, the phenotypic features of dyslipidemias have a similar clinical presentation, which is why biochemical diagnostic criteria have been developed, otherwise there would be no need for such criteria. Thus, there is no basis for such a statement by the authors.
The name "dysbetalipoproteinemia" stems from the abnormal gel electrophoresis migration pattern of VLDL, identified as a broad-β band, which is continuous from the β to pre-β levels (Fredrickson et al., 1967; Mahley and Rall, 1995). Currently, different approaches are used to identify FD and different criteria give different results, which is what the authors notice. Therefore, the authors should have focused on this and assessed the prevalence of FD in Russia according to the most optimistic and pessimistic criteria at least, but not to take only one.
A previous comment on the presentation of the study design in Figure 2 has been ignored by the authors. I believe this is a very important point and should be accepted by the authors and carefully reviewed. The current study design is very difficult to understand, especially from the steps described in the text.
In addition, the authors are recommended to highlight the problem of lipid-lowering therapy in Russia, as the authors were able to collect a large enough sample where almost all patients were not receiving lipid-lowering therapy, even in patients with signs of lipid metabolism disorders.
In general, the ambiguous research design of the manuscript can make it challenging for the reader to understand what the authors were trying to report. The integrity of the article remains questionable, on the one hand much has been done, but every part of it feels either unnecessary or understated. Patients are always bouncing between data, included here and excluded there, and it is very discouraging. The authors need to look at the integrity of the study and present it properly. I understand that the authors have done a lot and want to present as much as possible of everything that was done and found, and that would be fine if it fit into a single crystallized idea, but in the end the authors seem to present everything and nothing at the same time.
Minor:
The DOI of source No 17 is incorrect.
In Table S1, add version of Gnomad. Allele frequency (Gnomad) is obtained from exome or genome? Add date of access.
In Table S1, it is necessary to decipher all the abbreviations in the note.
In Table S2. There is no need to use abbreviations in the table, there is a lot of space.
